# Age-Dependent Changes in the Relationships between Traits Associated with the Pathogenesis of Stress-Sensitive Hypertension in ISIAH Rats

**DOI:** 10.3390/ijms241310984

**Published:** 2023-07-01

**Authors:** Dmitry Yu. Oshchepkov, Yulia V. Makovka, Mikhail P. Ponomarenko, Olga E. Redina, Arcady L. Markel

**Affiliations:** 1Federal Research Center Institute of Cytology and Genetics, Siberian Branch of Russian Academy of Sciences (SB RAS), Novosibirsk 630090, Russia; 2Department of Natural Sciences, Novosibirsk State University, Novosibirsk 630090, Russia

**Keywords:** principle component analysis, hypertension, stress reactivity, age-related difference, ISIAH rat strain

## Abstract

Hypertension is one of the most significant risk factors for many cardiovascular diseases. At different stages of hypertension development, various pathophysiological processes can play a key role in the manifestation of the hypertensive phenotype and of comorbid conditions. Accordingly, it is thought that when diagnosing and choosing a strategy for treating hypertension, it is necessary to take into account age, the stage of disorder development, comorbidities, and effects of emotional–psychosocial factors. Nonetheless, such an approach to choosing a treatment strategy is hampered by incomplete knowledge about details of age-related associations between the numerous features that may contribute to the manifestation of the hypertensive phenotype. Here, we used two groups of male F_2_(ISIAHxWAG) hybrids of different ages, obtained by crossing hypertensive ISIAH rats (simulating stress-sensitive arterial hypertension) and normotensive WAG rats. By principal component analysis, the relationships among 21 morphological, physiological, and behavioral traits were examined. It was shown that the development of stress-sensitive hypertension in ISIAH rats is accompanied not only by an age-dependent (FDR < 5%) persistent increase in basal blood pressure but also by a decrease in the response to stress and by an increase in anxiety. The plasma corticosterone concentration at rest and its increase during short-term restraint stress in a group of young rats did not have a straightforward relationship with the other analyzed traits. Nonetheless, in older animals, such associations were found. Thus, the study revealed age-dependent relationships between the key features that determine hypertension manifestation in ISIAH rats. Our results may be useful for designing therapeutic strategies against stress-sensitive hypertension, taking into account the patients’ age.

## 1. Introduction

Arterial hypertension is a polygenic disorder, the development of which is affected by numerous environmental and endogenous factors. Many risk factors and comorbidities of hypertension are known. The most common are diabetes mellitus, overweight, obesity, and chronic kidney diseases [1,2,3]. It is believed that some of these pathologies, such as chronic kidney disease, can be either a cause of a hypertensive state or a complication of uncontrolled hypertension [2]. Hypertension is possibly the most powerful modifiable risk factor of heart failure and other cardiovascular complications including myocardial infarction (and cerebrovascular diseases such as stroke). Chronic hypertension drives myocardial remodeling, resulting in hypertensive heart disease [4,5]. 

Despite major efforts of researchers and physicians to control hypertension, including treatment of hypertension in patients with comorbid conditions, numerous questions about the diagnosis, patient evaluation and monitoring, and the choice of a treatment strategy for hypertension remain highly relevant and not fully resolved [6,7]. 

By now, it is recognized that it is necessary to take into account different stages of the development of the disorder when diagnosing and choosing a strategy for treating hypertension [6]. It is assumed that at different stages of hypertension progression, various pathophysiological processes may play a key role in the manifestation of the hypertensive phenotype [8]. This notion raises questions about strategies for prescribing antihypertensive drugs depending on age, the disorder stage, and the presence of comorbidities [6,9,10,11]. It has been emphasized that strategies for the treatment of arterial hypertension in the elderly should take into consideration not only concomitant diseases but also the effects of emotiogenic psychosocial factors [12]. 

Understanding the importance of the assessment of age-related differences in functional expression of stressor-related negative affect [13,14] is undoubtedly important but is hampered by incomplete knowledge about the details of age-related associations between the numerous features that may contribute to the manifestation of the hypertensive phenotype. Conducting studies on the age-dependent component of stressor-related negative affect in human populations is difficult due to differences in study designs, populations, measurement, in the operational definition, and in the analytic approach, which lead to heterogeneity of findings [13]. Animal strains that model human diseases are often a successful alternative to population studies. 

ISIAH (inherited stress-induced arterial hypertension) rats are a proven model of a stress-sensitive form of hypertension and make it possible to investigate the physiological and molecular genetic mechanisms behind the development of a hypertensive state associated with increased stress reactivity under conditions of emotional stress [15,16,17,18]. 

Principal component analysis, which is based on correlation analysis, allows the assessment of the relationships between many traits and the identification of a set of main factors (principal components) that can contribute to the manifestation of certain phenotypic characteristics [19]. Previously, we have used principal component analysis to examine relationships among phenotypic characteristics in a group of 3-month-old male F_2_(ISIAHxWAG) hybrids obtained by crossing hypertensive ISIAH rats and normotensive WAG rats [20]. The use of the group of hybrid rats enabled us to obtain a full range of variation between normal and hypertensive phenotypes. In that paper, we analyzed 21 morphological, physiological, and behavioral traits, many of which show interstrain differences and are usually considered key features of the development of the hypertensive phenotype. Behavior was assessed in the open field test, which helps researchers to evaluate basic psychophysiological characteristics: the severity of fear and anxiety reactions, locomotor activity, and levels of exploratory and displacement activities [21]. In that research article, the first principal component was found to be associated with behavioral traits of F_2_(ISIAHxWAG) rats in the open field test. The basal blood pressure (BP) level contributed to the second and fourth principal components with the opposite sign. At the same time, the contribution to the second principal component was also made by weight traits: body weight and weights of peripheral organs that are targets of hypertension (kidneys, heart, and adrenal glands). The contribution to the fourth principal component, in addition to the basal level of BP, was made by all traits that are related to the stress-induced state of rats: BP under stress, adrenal weight, plasma corticosterone concentration under stress, and the defecation score in the open field test [20]. 

The purpose of the present work was to determine the relationships between the same traits in a group of older (six-month-old) males of F_2_(ISIAHxWAG) rat hybrids and to identify possible changes in these relationships during maturation of the animals, i.e., with the progression of the disorder.

## 2. Results

### 2.1. Age-Dependent Differences in Phenotypic Traits between Two Groups of F_2_(ISIAHxWAG) Hybrid Rats of Different Ages

A comparison of the phenotypic traits of F_2_(ISIAHxWAG) rats between age groups of 3–4 and 6–7 months (Table 1) uncovered many significant differences. Table 1 shows that as rats grow older, there is a statistically significant increase in the basal level of BP, but the increase in BP under restraint stress in the group of older rats is significantly weaker. Taking into account that the increase in plasma corticosterone concentration in the group of rats aged 6–7 months is also smaller as compared to 3–4-month-old animals, we propose that there is a decrease in stress responsiveness when the rats grow up.

Because body weight increases with age and absolute weight values of a target organ are expected to increase, only relative weights of the kidneys, heart, and adrenal glands were used in the further analysis. Such traits as BP under stress and the increase in BP under stress as well as the plasma corticosterone concentration under stress and the increase in plasma corticosterone concentration under stress are interdependent. Therefore, only traits representing stress-induced changes in their values were employed in the subsequent analysis because they reflect stress responsiveness in rats. The increase in corticosterone concentration under the influence of stressors is regarded as one of the key signs of adaptation of the body to the stressful conditions [22]; consequently, the relationships between plasma corticosterone concentration at rest (and its increase under stress) and both morphometric traits and behavioral traits were studied here. 

### 2.2. Analysis of Associations of BP Traits and of Weight Parameters with Plasma Corticosterone Concentration

Figure 1 shows the results of the examination of eight traits (measured in rats of both ages) by principal component analysis. The traits characterizing the weight characteristics of organs are mainly projected onto the horizontal axis (principal component 1); the vector of the absolute body weight and vectors of the relative weights of the adrenal glands, kidneys, and heart have opposite orientations. This finding has a natural explanation: the greater the body weight, the lower the relative weight of the organs. The body weight (a significant negative correlation) and relative weight of the heart and adrenal glands (significant positive correlations) produce the highest loading on the first principal component. Accordingly, the horizontal axis can be described as the axis associated with the weight characteristics of the experimental rats.

As for the second principal component, the main loadings on it are given by BP values: the basal-BP vector has a significant negative correlation with the second principal component, and the BP increase under stress (ΔBP) vector has the opposite direction and a positive correlation with the second principal component. Thus, the second principal component can be characterized as a factor that determines BP stress reactivity, and an increase in this factor is characteristic of the ISIAH rat strain.

Graphical representation of the contributions of traits to the principal components in the combined group of F_2_(ISIAHxWAG) hybrid male rats of different ages (Figure 1) clearly indicated the presence of a significant negative correlation between the second principal component and the age factor. The table of correlations of traits with principal components (Table 2) shows that the second principal component explains 19.32% of the variance of traits associated with age. Overall, the first two principal components account for almost 45% of the variance of the traits presented in Table 2.

Accordingly, the performed analysis suggests that it is the development of the hypertensive status that is the main age-dependent parameter in which the groups of F_2_(ISIAHxWAG) males differ. A positive correlation of age with the basal level of BP and a negative correlation of age with an increase in BP under stress indicate a decline in stress reactivity in the group of male hybrids aged 6–7 months compared with the young group of the animals.

For a more detailed assessment of the influence of the age factor on the variance of the studied parameters, an additional analysis was performed on each age group of rats separately.

The results from comparing the relationships of traits in each age group (Table 3) showed the same contribution of three weight parameters (body weight and relative weights of the heart and adrenal glands) to the first principal component. In both groups, the body weight inversely correlated with the relative weights of the adrenals and heart. The plasma corticosterone concentration in both groups of rats makes the greatest contribution to the second principal component, and both traits of BP make the largest contribution to the third principal component. Nevertheless, we were able to identify some age-dependent differences. The analysis suggested that in the young group of F_2_(ISIAHxWAG) hybrid rats, the assessed traits of BP, body weight, and relative weights of the target organs are not associated with the plasma corticosterone concentration and its increase under short-term restraint stress.

In comparison to the young group of rats, in male F_2_(ISIAHxWAG) hybrids at the age of 6–7 months, the contribution of the basal BP level to the first principal component is weaker (Table 3). In the second principal component, a contribution of two traits (plasma corticosterone concentration and the relative weight of the kidneys) emerged. In contrast to the young group of rats, in older rats, we in older rats a contribution of the increase in plasma corticosterone concentration under restraint stress (along with both traits of BP) to the third principal component and to the fourth principal component (along with the relative kidney weight). Therefore, in the adult group of the animals, in contrast to the young group, the trait “the increase in the plasma concentration of corticosterone under restraint stress” manifested an interdependence with both traits of BP and with the relative weight of the kidneys. It should be noted that the contribution of the trait “relative weight of the kidneys” to the second and fourth principal components was opposite.

Graphical representation of the data revealed that in the group of 3–4-month-old F_2_(ISIAHxWAG) rats, orientations of the vectors of plasma corticosterone concentrations and weight traits are almost orthogonal (Figure 2a). This state of affairs changed significantly by age 6–7 months, when almost all traits began to interact negatively with basal plasma corticosterone levels and positively with corticosterone gains (Figure 2b).

### 2.3. Analysis of the Associations of Behavior in the Open Field Test with BP and Hormonal Parameters

The analysis of behavior, BP, and hormonal parameters in the combined group of rats of both age ranges showed that six out of seven behavioral characteristics of rats in the open field test make a significant positive contribution to the first principal component, which is responsible for 27% of the total variance of the traits measured in this group (Figure 3, Table 4). A negative contribution to the first principal component was made by the latency period. Neither BP traits nor hormonal characteristics contributed significantly to the first principal component. Also unrelated to the first principal component is the defecation score in the open field test. Thus, the first principal component characterized the behavioral (largely exploratory) activity. The overall level of activity of rats in the open field test did not show a significant association with the age factor, with parameters of hypertensive status, or with glucocorticoid adrenal function.

The characteristics of the hypertensive status made the main contribution to the second principal component, which accounted for 15.4% of the total variance, with a positive contribution from basal BP and a negative contribution from the increase in BP under stress. Apparently, the higher the basal BP, the lower the amplitude of the increase in BP under stress. This pattern is possibly due to the tension of the depressor baroreflex with elevated basal pressure. Some characteristics of behavior also correlated with the second principal component: a negative contribution was noted for locomotor activity, and a positive contribution was noted for such parameters as displaced activity (grooming) and the latency period (Table 4). That is, in this case, we are dealing with parameters of behavior that characterize a conflict of several motivations: anxiety, fear, and exploration. “Age indicator” positively and significantly correlated only with the second principal component. Consequently, the second component, associated with the age of rats, can be characterized as a factor of behavioral caution (indecisiveness) and anxiety, which increases with age and with the development of persistent hypertension.

Characteristics of hormonal function—the basal plasma corticosterone level and its increase under stress—were found to make a significant contribution to the third principal component (11.28% of the total variance). By analogy with the basal and stress-induced rise in BP, the vectors of the basal and stress-induced increase in plasma corticosterone levels have mutually opposite directions: positive for the basal level and negative for the increase in plasma corticosterone under stress. Overall, the reciprocity between basal and stress-induced levels of corticosterone can be explained by negative feedback in the hypothalamic–pituitary–adrenal axis, and elevation of the basal level of the hormone should reduce the hormonal response to stress. It is possible that the negative feedback mechanism works more efficiently in young rats, although this hypothesis needs additional evidence. In addition, a significant positive contribution to the third principal component was detected for the defecation score. The association of this parameter in the coordinates of the third principal component with hormonal characteristics gives grounds to regard the defecation score as an indicator of emotionality and to describe the third principal component as a component associated with the magnitude of autonomic hormonal tension or stress.

The fourth principal component was also of interest. It proved to be positively loaded by such traits as increases in BP and plasma corticosterone under stress and by grooming (its frequency and duration) at the periphery of the open-field area. These data allowed us to associate this principal component with anxiety related to stress reactivity (Table 4).

A separate analysis of rats of different ages almost did not alter the structure of the first principal component. In both young and adult rats, the main loading on this component was provided by behavioral traits. The only difference was that in young rats, the negative contribution of the latency period did not reach statistical significance. Therefore, the first principal component in both age groups, just as in the combined group, is a factor of the overall behavioral activity of rats (Table 5). Meanwhile, the set of features associated with subsequent principal components changed rather strongly. Thus, the second principal component in young rats is essentially defined by parameters of the hormonal status: there is a significant positive correlation with the basal level of plasma corticosterone and a negative correlation with the upregulation of corticosterone under stress. On the other hand, reciprocity of the variances of these parameters persisted. In adult rats, the basal level of corticosterone did not contribute to the second principal component, but there remained a significant contribution from the increase in the level of plasma corticosterone under stress. This result confirms the abovementioned hypothesis about the weakening of the negative feedback in the hypothalamic–pituitary–adrenal axis in adult rats compared with young ones.

Furthermore, in adult rats, a significant positive contribution to the second principal component was made by basal BP together with the frequency and total duration of grooming in the open-field area. In young rats, no such correlations were detectable. Obviously, in adult rats, along with an increase in basal BP, the level of behavioral manifestations of anxiety goes up.

The structure of the third principal component is virtually the same between young and adult rats. In both age groups, the third principal component was found to be loaded with BP traits and could be characterized as a principal component associated with stress-dependent enhancement of BP. Moreover, in young and adult rats, the third principal component included grooming parameters, thereby revealing an association of the BP response to stress with behavioral status of the rats.

The fourth principal component indicated that in young rats, anxiety behavior might be somewhat unrelated to BP or plasma corticosterone levels. In adult rats, other traits contributed to the fourth principal component: a positively correlating increase in BP under stress and rearing duration and a negatively relating basal plasma corticosterone level.

The fifth principal component in both groups was shown to be represented by the defecation score in the open field test; this parameter had a very large and significant contribution only to this principal component, thereby enabling us to define the defecation score as a scale of “emotionality”. In contrast to the adult group of rats, in young rats, the latency period also made a significant contribution to the fifth principal component. Both traits in young rats did not depend on the level of BP or glucocorticoid function of the adrenal glands. In adult rats, however, there was an inverse relationship between the defecation score and the basal level of corticosterone.

Thus, this analysis allows us to conclude that in the young group of F_2_(ISIAHxWAG) hybrid male rats, the behavioral traits assessed in the open field test do not explicitly depend on the plasma corticosterone concentration and on its increase under short-term restraint stress. In rats of the older group, the glucocorticoid function of the adrenal glands can modulate a number of behavioral traits related to emotionality. It is apparent that in contrast to young rats—where only one principal component is loaded with both traits of BP—in rats of the adult group, several principal components that reflect behavior are loaded with BP parameters in different combinations. A graphical representation of the contributions of behavioral traits to the first two principal components in groups of male F_2_(ISIAHxWAG) hybrids of different ages is presented in Figure 4.

## 3. Discussion

The development of hypertension is accompanied by alterations to the functioning of numerous physiological systems. In this study, we examined age-dependent changes in the relationships between traits associated with the pathogenesis of stress-sensitive arterial hypertension as well as with the behavioral traits of F_2_(ISIAHxWAG) male rats in the open field test. In our work, the use of F_2_ hybrids obtained by crossing hypertensive ISIAH rats with normotensive WAG rats can be considered a population model in which the diversity of phenotypes is provided by different combinations of alleles associated with both hypertension and high stress reactivity, as assessed after short-term restraint stress, which can be regarded as emotional stress. The analysis of two groups of rats of different ages made it possible to identify age-dependent phenotypic characteristics associated with the development of stress-sensitive hypertension as well as to evaluate alterations of associations between traits as the animals grew older. In both series of analyses performed on two distinct sets of traits (Table 2 and Table 4), a significant (FDR < 5%) relationship with age was found only for the second principal component. In both cases, it was demonstrated that the development of a hypertensive status is the main age-dependent parameter in which the groups of F_2_(ISIAHxWAG) males differ. This result indicates the adequacy of the chosen model for identifying associations related specifically to the hypertensive phenotype. The observed increase in BP with age is well known and points to aggravation of the development of hypertension. The vascular resistance to blood flow is enhanced due to many factors changing over time [23]. Our experiment showed that the presence of a positive relationship between age and basal BP is accompanied by a negative relationship between age and the increase in BP under stress. These results suggest that there was a significant decline of stress reactivity in the group of male F_2_ (ISIAHxWAG) hybrids at the age of 6–7 months compared with the group of rats aged 3–4 months. These data are in good agreement with research on the age-dependent differences in stress responsiveness among spontaneously hypertensive rats (SHRs). The hypothalamic–pituitary–adrenocortical response to acute stress is markedly enhanced in SHRs during early development of hypertension and does not differ in older SHRs from that in normotensive Wistar–Kyoto (WKY) and Wistar rats [24]. A decrease in stress reactivity of the cardiovascular function with ageing has been associated with a decline in β-adrenergic responsiveness with age [25]. Additionally, an impaired ability to recover from exposure to stressful stimuli in the elderly may be a consequence of chronically elevated glucocorticoid levels, which are characteristic of many age-related disorders, including hypertension [26]. The results of our experiment are in good agreement with these notions and indicate in both age groups of rats the presence of a negative correlation between the concentration of corticosterone in the blood plasma at rest and its increase under restraint stress.

Cortisol (corticosterone in rats) is a major stress-related glucocorticoid hormone. Its activation under stress contributes to the restoration of homeostasis and exerts effects on cardiovascular function, metabolism, muscle function, behavior, and the immune system [22]. Elevation of the cortisol level during stress helps to restore the functioning of various organ systems. Cortisol concentration in bodily fluids is considered a reliable indicator of stress. Cortisol reactivity is regarded as one of possible mechanisms through which psychosocial stress may influence the risk of hypertension [27].

According to our principal component analysis, in the young group of animals in our experiment, the traits of plasma corticosterone were not directly related to any of the traits being analyzed (Table 3 and Table 5). These findings are consistent with a report that in infants, the cortisol level, its reactivity to stress, and behavior are unrelated [28].

By contrast, in our group of rats aged 6–7 months, straightforward relationships were detected between traits of plasma corticosterone concentration and other assessed traits. Moreover, the increase in corticosterone concentration contributed to the third principal component, in addition to the positive contribution of the basal level of BP and the negative contribution of the increase in BP under stress (Table 3).

There is an opinion that low reactivity of BP to stress indicates the presence of an existing disease or disorder. The blunted cardiovascular and cortisol reactions to acute psychological stress have been linked to depression, obesity, behavioral disorders, and poorer cognitive ability, as reviewed in refs. [14,29]. It has been hypothesized that the detection of a blunted stress response may be informative for identifying patients who require closer medical attention [29]. One of the reasons for the blunted stress reactivity is genetic polymorphisms [30,31]. Because the ISIAH rat strain has been created by selection for a sharp increase in BP under short-term restraint (emotional) stress, we can theorize that genetic features of ISIAH rats may contribute to a decline in stress reactivity during the progression of the disorder in question. This supposition is supported by a report that many genes whose transcription differs between ISIAH and normotensive control rats are associated with the stress response [32,33,34,35,36]. This observation suggests that ISIAH rats—even at rest—are in a state of chronic functional stress.

We have previously shown that in adult ISIAH rats (4–5 months old), just as in normotensive WAG rats, significant elevation of the plasma corticosterone concentration takes place 15 min after the onset of restraint stress. During the next 15 min, however, the level of corticosterone in normotensive rats does not increase, while in hypertensive ISIAH rats, it continues to rise sharply and becomes significantly higher than that in control rats [17]. These differences signify an exaggerated stress response in hypertensive ISIAH rats compared to normotensive control WAG rats.

This phenomenon may be not only be characteristic of ISIAH and WAG rats, as evidenced by results of studies on the activation of immediate early genes described below. A number of studies on the hypothalamus of hypertensive and normotensive rats show differences in kinetics of transcription of the *Fos* gene, which is a marker of neuronal activation under stress. In normotensive Sprague–Dawley rats exposed to restraint stress, the peak of *Fos* gene activation in the paraventricular nucleus of the hypothalamus is observed 15 min after the onset of stress, and after 1 h, it returns to the control level [37]. In the research on *Fos* gene activation in the hypothalamus under the action of short-term restraint stress in hypertensive rats, somewhat different results were obtained. The peak of transient upregulation of *Fos* gene transcription in the hypothalamus of borderline hypertensive male rats occurred after 30 min of immobilization stress and returned to baseline only after 3 h [38]. Another report, on *Fos* gene activation in the paraventricular nucleus of the hypothalamus of male hypertensive SHRSP (spontaneously hypertensive, stroke-prone) rats under restraint stress, has also shown a peak of *Fos* gene activity at 30 min after the onset of stress [39]. As the hypertensive SHRSP rats mature and the pathophysiological state worsens, the *Fos* gene is activated under short-term (30 min) stress to a greater extent than in normotensive control animals [40]. Accordingly, those researchers proposed that the prolonged activation of the *Fos* gene may be a consequence of the pathophysiological state of rats associated with features of the manifestation of hypertensive status. 

In the pathophysiological state of the body, a prolonged stress response, including prolonged elevation of BP following stress, is linked with the inability to shut off stress responses essential for adaptation, homeostasis, and survival, even under conditions in which these responses are no longer required (e.g., when a stressor is removed or a stressful situation is over) [14,41]. The impaired recovery from stress (magnitude and/or duration of a stress response) is connected with the development or worsening of a disease [14]. Based on the foregoing, it can be hypothesized that the processes of hypertension progression in ISIAH rats and the increase in the duration of recovery from stress may be related.

In our experiment, we also revealed age-related correlations with other traits associated with the manifestation of hypertensive status. In contrast to the young group of F_2_(ISIAHxWAG) hybrid rats, in which all the assessed parameters do not explicitly depend on the concentration of plasma corticosterone and its increase under short-term restraint stress, in adult rats, both plasma corticosterone traits (along with other features) contribute to several principal components (Table 3 and Table 5). These results support the notion that during aging, there are alterations to the activity of various endocrine systems, including changes in hormonal secretion and modulation of feedback sensitivity [42]. Our findings are in good agreement with the known concept that there are several stages in the development of hypertension [8] as well as with the reports that the outcome of pharmacotherapy of hypertension may depend on age, i.e., on the stage of disorder development [43]. The fact that a phenotype controlled by many genes can be influenced by different genetic loci at different ages has been demonstrated both in research on the molecular genetic mechanisms of the development of the pathology in ISIAH rats [44,45,46] and in articles on rats that are models of other types of hypertension [47,48,49] as well as in a study on anthropometric parameters (height, weight, body mass index, and systolic BP) in humans [50].

One of the advantages of the approach to data analysis presented here is that it reveals a dual nature of features by revealing a set of features involved in direct and inverse relationships. This information may be useful for interpreting genetic mapping data on these traits and for identifying candidate genes at common genetic loci involved in the control over these traits.

Our analysis suggests that the development of a hypertensive status in ISIAH rats is accompanied by an increase in anxiety (Table 4). Furthermore (Table 5), one can see that in the young group of rats of F_2_(ISIAHxWAG) hybrids, the basal value of BP and the BP increase under short-term restraint stress contribute only to one (second) principal component. By contrast, in adult rats, the BP traits (along with other traits) contribute to several principal components; these other traits include basal levels and an increase in the corticosterone concentration, thereby pointing to the emergence of a relationship between behavior and hormonal and hypertensive status in rats with age. This result is in good agreement with conclusions from an analysis of population data suggesting that the elderly with hypertension are more likely to have depression and anxiety [51]. These mental problems are associated with a lower quality of life, a lower rate of treatment compliance, and higher mortality among elderly individuals [51]. The relevance of this problem is also emphasized by the fact that hypertension is also considered a risk factor for dementia [52]. 

Although some results of our study and the conclusions drawn are consistent with the data obtained earlier both in studies on other strains of hypertensive rats and in human population studies, we report some age-dependent associations for the first time. Our work confirms the complex nature of associations between key hypertension-related traits and identifies combinations of positive and negative relationships between some of them. Our findings suggest that ISIAH rats are an adequate model for identifying molecular genetic mechanisms linked with the development of anxiety during aging and the exacerbation of the pathogenesis of the stress-sensitive form of hypertension.

As with any experimental work, our study has several limitations. Certainly, one limitation is that we analyzed only males. In addition, it should be mentioned that the patterns observed on model animals cannot be considered an exact replica of the processes in the human body; nevertheless, no one doubts that the molecular and physiological mechanisms and their regulation are largely similar between humans and rodents. 

Our paper does not contain real clinical evidence but does offer avenues and clear guidelines for further research because we determined the parameters that should be examined in human population studies on the nature of the stress-sensitive form of hypertension and on age-dependent features of its manifestation.

## 4. Materials and Methods

### 4.1. Animals

We used two groups of F_2_(ISIAHxWAG) hybrid males at ages 3–4 (*n* = 103) and 6–7 months (*n* = 126). The hybrids were obtained by crossing ISIAH/Icgn rats, which are a breeding model for inherited stress-induced hypertension, with normotensive WAG/GSto-Icgn rats (Wistar Albino Glaxo). The rats were kept under standard conditions in a vivarium at the Institute of Cytology and Genetics SB RAS (Novosibirsk, Russia), with free access to feed and water. Experiments were conducted in accordance with the International Standards for the Care and Use of Laboratory Animals and were approved by the Bioethics Committee of the Institute of Cytology and Genetics for the work with experimental animals (protocol No. 69 of 20 January 2021).

### 4.2. Measurements of BP and Other Traits 

Interstrain differences in the evaluated traits of ISIAH and WAG rats and of their F_1_(ISIAHxWAG) hybrids and F_2_(ISIAHxWAG) hybrids at the age of 3–4 months are given in ref. [20]. A graphical abstract summarizing the time course of the experimental data collection is presented in Figure 5.

BP measurements in rats were performed by the tail-cuff method (described in detail elsewhere [17]). Briefly, basal values of BP were measured in anesthetized rats to exclude the stress associated with the tail-cuff procedure. To measure the stress-induced BP and plasma corticosterone response, rats were placed in a small wire-mesh cylinder cage for 0.5 h. BP after stress was measured without anesthesia. Blood was collected from the tail vein immediately after restraint. Then, the rats were returned to their home cages. One week later, the animals were rapidly decapitated. Samples of plasma were assayed for the basal corticosterone level. The target organs of hypertension (heart, kidneys, and adrenals) were excised and weighed. Plasma corticosterone levels were assayed by the competitive protein-binding radioassay technique [53]. Procedures for measuring BP and blood sampling for measuring the concentration of corticosterone in plasma were carried out at the same time of day, namely from 13:00 to 15:00 h local time.

### 4.3. Parameters of Rat Behavior

The parameters of rat behavior analyzed in the open field test were selected on the basis of previous studies because these parameters were the most informative [19]. Behavioral characteristics were evaluated in an open-field arena with automatic registration as described earlier [46]. The test lasted 6 min. The following behavioral characteristics of rats were analyzed: (1) locomotor activity in the first minute of the first trial of the open field test (number of crossed squares); (2) locomotor activity on the periphery of the open-field area (the sum of crossed squares for four trials); (3) grooming on the periphery of the open-field area (number of grooming acts in four trials); (4) grooming time, s (sum over four trials); (5) standing up on the hind legs (rearing) on the periphery of the open-field area (number of rearings in four trials); (6) rearing time, s (sum over four trials); (7) defecation (number of boluses in four trials); (8) latency period, s (time from placement on the platform to the start of locomotion, total for four trials). Rats were tested on 4 consecutive days (four trials). The sum of values from the four trials gave an integral assessment of behavior, which largely eliminated the influence of random factors on such a labile indicator as behavior. The locomotor activity in the first minute of the first trial was analyzed separately from the general locomotor activity because the locomotor activity of the rat in the first minute mainly corresponds to the reaction of avoiding an unfamiliar environment, in contrast to the locomotor activity at the end of the first and subsequent trials, which contributes mostly to the principal component associated with exploratory behavior [19].

### 4.4. Statistical Analysis

Statistical analysis was performed using PAST software version 4.03 (PAST 4.03) [54]. Several traits were log-transformed by taking the natural logarithm to eliminate the skewness and kurtosis. Results of our previous study have shown that behavioral traits of rats in the open field test and morphometric traits contribute to different principal components [20]. Taking this into account, in the present work, the analyses of behavioral and morphometric traits were carried out separately. Principal component analysis was performed in two versions: (1) separately for 3–4- and 6−7-month-old rats; (2) jointly for the two age groups, where the parameter “age indicator” served as an independent categorical (dummy) variable in the calculation of correlations with principal components. The calculation of the significance of the Pearson correlation coefficient (*r*) was conducted via the Benjamini–Hochberg correction for multiple comparisons. The data were considered significant at FDR < 5%.

## 5. Conclusions

BP and an increase in BP under stress are the traits that we consider the main endpoints in this paper. Nonetheless, they are formed by combined action of many endogenous and exogenous factors. In turn, high BP may have an impact on the same factors (a feedback loop). Elevation of BP and an increase in the concentration of plasma corticosterone during stress are traits that characterize the response to stress. Weakening of the response to stress may be due to self-regulation of stress response systems: an increase in basal function automatically reduces the response to an additional stimulus, thereby leading to a state of chronic stress that is characterized by a persistent increase in basal BP and a rise of basal plasma corticosterone concentration. In our work, the use of the ISIAH rat strain, which simulates the stress-sensitive form of arterial hypertension, enabled us to analyze the traits of emotionality that rats exhibit in the open field test. Our data clearly show that the development of hypertensive status in F_2_(ISIAHxWAG) rats is accompanied by the enhancement of anxiety. The study revealed age-dependent relationships between key features and their changes with age and worsening of the hypertensive state. In particular, it was demonstrated that the concentration of plasma corticosterone at rest and its increase under short-term restraint stress in a young group of rats does not have a straightforward relationship with other assessed traits. By contrast, in older animals, such associations were found. Because emotion-related brain activity has been reported to be an important factor for the development of hypertension in humans [55], we hope that our analysis of relationships between the investigated traits (which are associated with a genetically determined hypertensive phenotype and high stress reactivity under conditions of short-term restraint (emotional) stress) will be useful for the design of strategies for the treatment of hypertension, taking into account the age of the patients.

## Figures and Tables

**Figure 1 ijms-24-10984-f001:**
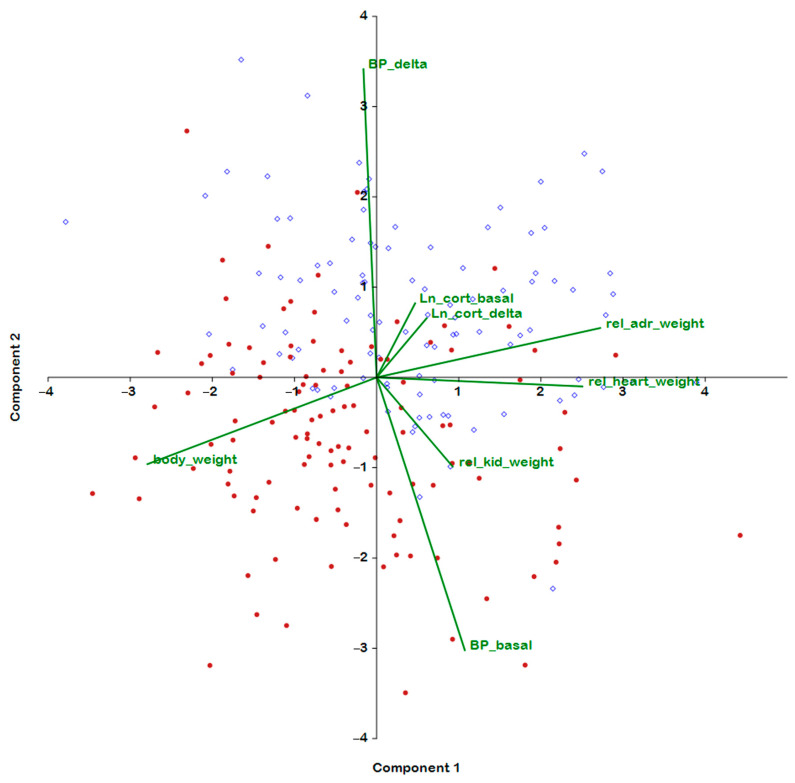
Principal component analysis performed on the combined group of male F_2_(ISIAHxWAG) hybrids of different ages. Blue dots: 3–4-month-old rats; red dots: 6–7-month-old rats.

**Figure 2 ijms-24-10984-f002:**
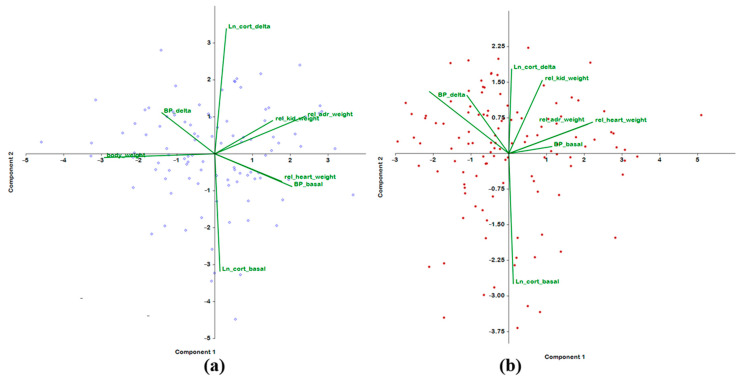
Principal component analysis in groups of F_2_(ISIAHxWAG) hybrid males of different ages: (**a**) 3–4 months old; (**b**) 6–7 months old.

**Figure 3 ijms-24-10984-f003:**
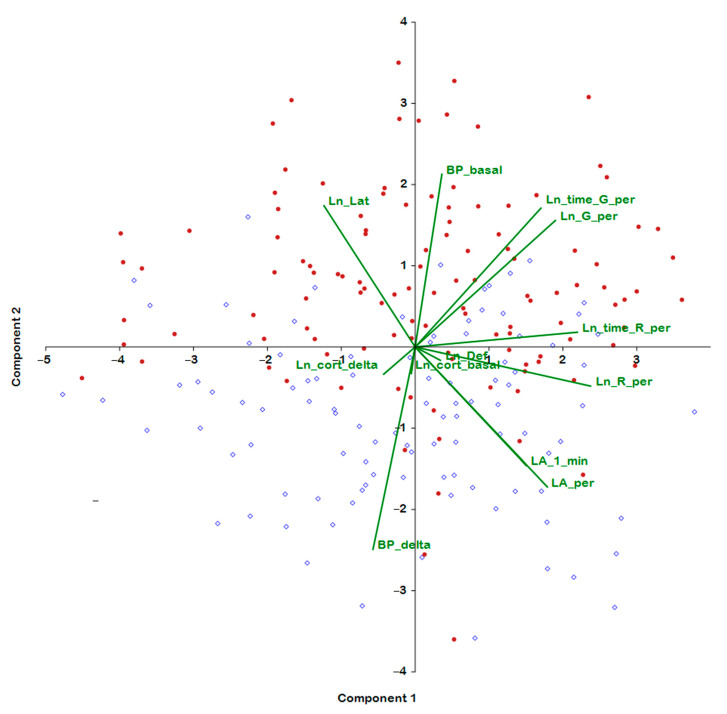
Principal component analysis (of behavioral traits) performed on the combined group of F_2_(ISIAHxWAG) hybrid males of the two age ranges. Blue dots: 3–4-month-old rats; red dots: 6–7-month-old rats.

**Figure 4 ijms-24-10984-f004:**
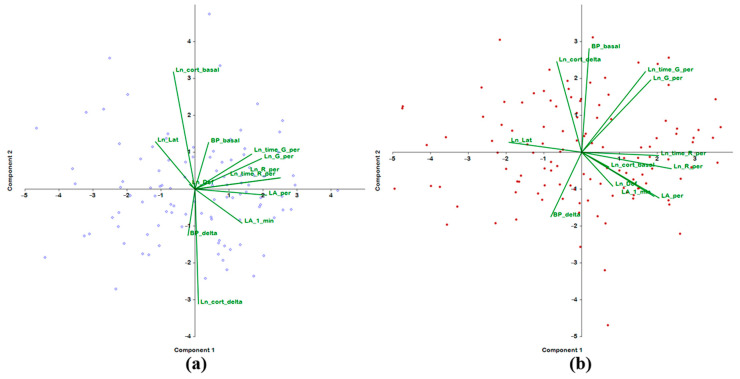
Principal component analysis of behavioral traits in groups of F_2_(ISIAHxWAG) hybrid males of different ages: (**a**) 3–4 months old; (**b**) 6–7 months old.

**Figure 5 ijms-24-10984-f005:**
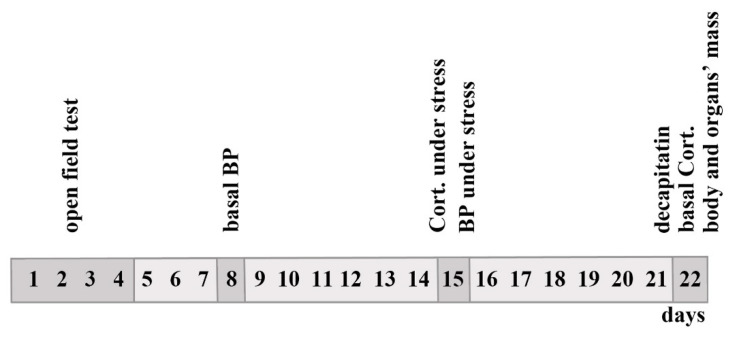
An outline of experimental data collection. BP: blood pressure; Cort.: corticosterone. Days 1–4: open field test; day 8: basal BP measurement; day 15: BP measurement under stress and blood collection for corticosterone quantification under stress; day 22: decapitation, blood collection for basal corticosterone quantification, and body and organ weight measurement.

**Table 1 ijms-24-10984-t001:** Phenotypic traits (mean ± SE) in the two groups of F_2_(ISIAHxWAG) hybrid rats of different ages.

Trait	F_2_(ISIAHxWAG) (3–4 Months Old) *n* = 101	F_2_(ISIAHxWAG) (6–7 Months Old) *n* = 118
Basal BP, mmHg	158.6 ± 1.6 ** ^xxx^	164.3 ± 1.7 ^xx^
BP under stress, mmHg	189.9 ± 1.8 ***	172.2 ± 1.8
Increase in BP during stress, mmHg	31.3 ± 2.0 ***	7.9 ± 1.9
Plasma corticosterone concentration at rest, µg/100 mL	3.38 ± 0.35 ** ^xxx^	1.92 ± 0.17 ^xxx^
Plasma corticosterone concentration during stress, µg/100 mL	31.6 ± 0.53 ***	25.5 ± 0.93
Increase in plasma corticosterone concentration under stress, µg/100 mL	28.2 ± 0.64 ***	23.7 ± 0.95
Body weight, g	253.9 ± 3.52 ***	297.4 ± 3.9
Adrenal weight, mg	36.0 ± 0.47 ***	39.5 ± 0.54
Adrenal weight/body weight, mg/100 g body weight	14.4 ± 0.23 **	13.4 ± 0.2
Kidney weight, g	1.52 ± 0.02 ***	1.87 ± 0.03
Kidney weight/body weight, g/100 g body weight	0.60 ± 0.006	0.63 ± 0.005
Heart weight, g	0.95 ± 0.016 ***	1.09 ± 0.01
Heart weight/body weight, g/100 g body weight	0.38 ± 0.0035	0.37 ± 0.003
Locomotor activity in the first minute of the first trial of the test, crossed squares	51.4 ± 2.3	47.4 ± 1.9
Locomotor activity on the periphery of the open-field area, crossed squares	319 ± 14.9 ***	241.3 ± 10.1
Grooming at the periphery of the open-field area	3.2 ± 0.36 ***	5.4 ± 0.5
Time of grooming at the periphery of the open-field area, s	43.4 ±0.52 ***	94.0 ±10.2
Rearings at the periphery of the open-field area	19.0 ± 1.2	19.2 ± 0.9
Time of rearing at the periphery of the open-field area, s	57.2 ± 3.8 ***	75.7 ± 4.0
Defecation	13.0 ± 0.76	13.2 ± 0.6
Latency period, s	40.0 ± 2.1 ***	84.5 ± 6.2

** *p* < 0.01 and *** *p* < 0.001 for the comparison between F_2_(ISIAHxWAG) rats at the age of 3–4 months and F_2_(ISIAHxWAG) rats at the age of 6–7 months; ^xx^
*p* < 0.01 and ^xxx^
*p* < 0.001 as compared to the stress group (Student’s *t* test).

**Table 2 ijms-24-10984-t002:** The table of correlations of traits with principal components in the combined group of male F_2_(ISIAHxWAG) hybrid rats of different ages.

Trait	PC 1	PC 2	PC 3	PC 4	PC 5
BP_basal	0.313	**−0.765**	0.029	−0.339	0.184
BP_delta	−0.047	**0.865**	0.178	0.114	0.213
Ln_cort_basal	0.136	0.208	**−0.808**	−0.193	0.269
Ln_cort_delta	0.177	0.166	**0.723**	**−0.482**	0.274
body_weight	**−0.813**	−0.243	0.216	0.103	0.188
rel_adr_weight	**0.793**	0.139	0.163	−0.058	**−0.428**
rel_kid_weight	0.270	−0.250	0.222	**0.800**	0.116
rel_heart_weight	**0.729**	−0.025	−0.045	0.189	**0.450**
age_indicator	−0.287	**−0.559**	−0.003	0.311	−0.094
% of variance	25.55	19.33	16.65	13.58	8.26

Significant Pearson correlation coefficients (false discovery rate (FDR) < 5%) are boldfaced.

**Table 3 ijms-24-10984-t003:** The table of correlations of traits with principal components in groups of F_2_(ISIAHxWAG) hybrid males at ages 3–4 and 6–7 months.

3–4 Months Old					
Trait	PC 1	PC 2	PC 3	PC 4	PC 5
BP_basal	**0.606**	−0.227	**−0.540**	−0.006	0.081
BP_delta	−0.416	0.284	**0.713**	−0.143	−0.151
Ln_cort_basal	0.041	**−0.814**	0.313	−0.241	0.087
Ln_cort_delta	0.091	**0.867**	−0.188	0.042	−0.083
body_weight	**−0.869**	−0.026	−0.126	0.135	0.104
rel_adr_weight	**0.709**	0.260	0.212	**−0.516**	−0.147
rel_kid_weight	0.454	0.230	0.426	0.256	**0.701**
rel_heart_weight	**0.526**	−0.191	0.332	**0.603**	−0.440
% of variance	28.64	21.31	16.07	9.91	9.51
**6–7 Months Old**					
**Trait**	**PC 1**	**PC 2**	**PC 3**	**PC 4**	**PC 5**
BP_basal	0.411	0.044	**0.761**	−0.246	0.117
BP_delta	−0.400	0.370	**−0.643**	0.272	0.243
Ln_cort_basal	0.043	**−0.830**	−0.058	0.095	**0.486**
Ln_cort_delta	0.027	**0.539**	**0.499**	**0.503**	0.379
body_weight	**−0.758**	0.394	0.288	−0.155	−0.030
rel_adr_weight	**0.756**	0.187	−0.154	0.335	−0.290
rel_kid_weight	0.320	**0.466**	−0.351	**−0.618**	0.300
rel_heart_weight	**0.801**	0.200	−0.167	0.024	0.148
% of variance	27.77	19.58	18.78	11.43	8.12

Significant Pearson correlation coefficients (FDR < 5%) are highlighted in bold.

**Table 4 ijms-24-10984-t004:** The table of correlations between behavioral traits and principal components in the combined group of male F_2_(ISIAHxWAG) hybrid rats of different ages.

Trait	PC 1	PC 2	PC 3	PC 4	PC 5
BP_basal	0.131	**0.581**	−0.376	−0.214	**0.539**
BP_delta	−0.207	**−0.680**	0.150	**0.428**	−0.240
Ln_cort_basal	−0.019	−0.089	**0.594**	−0.303	0.316
Ln_cort_delta	−0.155	−0.091	**−0.553**	**0.629**	0.259
LA_1_min	**0.545**	−0.398	0.110	−0.031	0.308
LA_per	**0.649**	**−0.470**	−0.239	−0.207	−0.003
Ln_G_per	**0.689**	**0.426**	0.280	**0.439**	−0.047
Ln_R_per	**0.862**	−0.132	−0.197	−0.179	−0.207
Ln_time_G_per	**0.618**	**0.467**	0.320	**0.468**	−0.040
Ln_time_R_per	**0.797**	0.050	−0.229	−0.192	−0.344
Ln_Def	0.124	−0.045	**0.435**	−0.042	0.127
Ln_Lat	**−0.448**	**0.475**	−0.020	−0.179	**−0.566**
age_indicator	0.086	**0.577**	0.039	−0.181	−0.280
% of variance	27.11	15.36	11.28	10.62	9.26

Significant Pearson correlation coefficients (FDR < 5%) are boldfaced.

**Table 5 ijms-24-10984-t005:** The table of correlations of behavioral traits with principal components in groups of F_2_(ISIAHxWAG) hybrid males at ages 3–4 and 6–7 months.

3–4 Months Old					
Trait	PC 1	PC 2	PC 3	PC 4	PC 5
BP_basal	0.139	0.318	**−0.632**	0.439	0.182
BP_delta	−0.071	−0.316	**0.719**	−0.330	0.013
Ln_cort_basal	−0.221	**0.801**	0.153	−0.201	0.101
Ln_cort_delta	0.035	**−0.785**	−0.071	0.397	−0.127
LA_1_min	**0.480**	−0.232	0.059	−0.207	0.128
LA_per	**0.733**	−0.041	−0.139	−0.411	−0.031
Ln_G_per	**0.681**	0.209	0.365	**0.515**	−0.168
Ln_R_per	**0.873**	0.077	−0.117	−0.249	0.011
Ln_time_G_per	**0.580**	0.239	**0.495**	**0.512**	−0.144
Ln_time_R_per	**0.852**	0.076	−0.092	−0.242	−0.007
Ln_Def	−0.057	0.032	0.388	0.149	**0.790**
Ln_Lat	−0.404	0.323	0.208	−0.106	**−0.502**
% of variance	27.48	14.45	13.02	11.61	8.35
**6–7 Months Old**					
**Trait**	**PC 1**	**PC 2**	**PC 3**	**PC 4**	**PC 5**
BP_basal	0.069	**0.676**	**−0.534**	−0.207	−0.081
BP_delta	−0.288	−0.421	**0.548**	**0.474**	−0.058
Ln_cort_basal	0.254	−0.097	0.216	**−0.628**	**−0.465**
Ln_cort_delta	−0.233	**0.591**	−0.063	0.355	0.063
LA_1_min	**0.675**	−0.288	0.025	−0.257	−0.082
LA_per	**0.723**	−0.298	−0.368	0.181	−0.007
Ln_G_per	**0.645**	**0.471**	**0.516**	0.020	0.079
Ln_R_per	**0.837**	−0.108	−0.179	0.340	−0.100
Ln_time_G_per	**0.596**	**0.526**	**0.521**	0.016	0.101
Ln_time_R_per	**0.718**	−0.022	−0.213	**0.469**	−0.099
Ln_Def	0.290	−0.221	−0.048	−0.303	**0.822**
Ln_Lat	**−0.679**	0.065	−0.069	0.176	0.017
% of variance	30.97	14.40	11.62	11.21	7.90

Significant Pearson correlation coefficients (FDR < 5%) are highlighted in bold.

## Data Availability

Not applicable.

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
