# Peer review of "Age-Dependent Changes in the Relationships between Traits Associated with the Pathogenesis of Stress-Sensitive Hypertension in ISIAH Rats"

_ijms, 2023, doi:10.3390/ijms241310984_

Round 1

Reviewer 1 Report

comments and suggestions are in attached form

Author Response

Age Dependent Changes in Relationships between Traits Associated with the Pathogenesis of Stress Sensitive Hypertension in ISIAH (Inherited Stress-Induced Arterial Hypertension) Rats. The authors wanted to group many dependent variables that are associated with the influence of numerous environmental and endogenous factors. The experimental model used are ISIAH rats, a well-known rat’s model in which they used the principal component method to analyze the relationships between phenotypic characteristics in a group of young (at the age of 3 months) male F2(ISIAHxWAG) hybrids obtained by crossing hypertensive ISIAH and normotensive WAG rats. The use of a group of hybrid rats allowed the authors to obtain the full range of variability between normal and hypertensive phenotypes. Results of the work are clear and well described. The results probably are not sound as a new knowledge but they can analyze the different variables simultaneously, and give a strong evidence for the majority of diseases in which the organism is responding as a whole and are many organs and system involved.

Answer: The authors thank the reviewer for the high evaluation of our work and comments, and suggestions that allowed us to improve the text of the manuscript. For the convenience of reviewers, all corrections made to the text of the manuscript are shown in red. As recommended by the reviewers, the text of the manuscript has been edited by a professional translator. The language certificate is enclosed.

In methods, section 4.2 is difficult to this reviewer to follow the time course for sampling, as an example: BP was measured at day 8 and a second BP was done at 15 days the day after the blood sampling for corticosterone.

  • The question is to know when stress was done? -It was only one session? – yes, it was only one session of stress. We have added text to the legend of Figure 5 explaining the steps for collecting experimental data.
  • How many days after the stress was done, occurred the blood sampling? - Blood was collected from tail vein immediately after the completion of the restriction stress procedure (lines 530-531).
  • Why in one condition the BP in rats was done in anaesthetized rats any in other no? - Anesthesia was used to avoid the stress associated with the tail-cuff procedure (lines 527-528), i.e. to get basal blood pressure values. When measuring blood pressure under stress, it is not required to avoid the stress associated with the tail-cuff procedure, so anesthesia is not used.
  • All of these doubts can be solve by including more information in figure 5.

Answer: The authors are grateful to the reviewer for pointing out that Figure 5 is not completely clear. We have added text to the legend of Figure 5 explaining the steps for collecting experimental data, so that readers do not get confused about the time course of the experimental data collection. In addition, we have combined subsections 4.2 and 4.3 so that the information in Figure 5 is read in conjunction with the text that was presented in subsection 4.3.

  • The authors have a reason why they use male rats in the study and not females? This is taking in consideration that the response to corticoids’ in male compared to female are different
  • Answer: Initially, when creating the strain, both males and females were tested. It has been shown that females have some features of the development of hypertension. Therefore, studying the physiological and molecular genetic mechanisms of the development of hypertension should be carried out taking into account the sex of the animals. Due to the fact that work on both females and males requires twice as much labor and financial costs, the main work of the group is carried out only on males. This limitation is highlighted at the end of the discussion (Line 494).

It was interesting to follow the central emotional modifications and the correlation with stress at the periphery. This is a strength of the work

Answer: Thank you

Please define ISIAH and SHRSP the first time is used

Answer: done -  lines 75 (for ISIAH) and line 431 (for SHRSP)

Reviewer 2 Report

This is an experimental research work on animals using Two groups of male F2(ISIAHxWAG) hybrids of different ages. A statistical analysis of principal component data science is carried out, analyzing the relationships of morphological, physiological and behavioral traits. It has been shown that the development of stress-sensitive hypertension, accompanied by a persistent increase in basal blood pressure and an increase in the basal concentration of plasma corticosterone, is accompanied by a decrease in response of these parameters to stress and by an increase in anxiety. The study revealed age-dependent relationships between key features and their changes with age and worsening of the hypertensive state.

Regarding the manuscript in detail:

The title is clear, concise, brief and well describes the work and model used.

The abstract is clear, but it should be corrected to be structured, respect the different sections of the manuscript and provide some numerical data on results and levels of statistical significance.

The introduction is somewhat extensive, but it still meets the requirements of adequately explaining the theoretical foundations of the research, raising the doubts that lead to this work, and laying the foundations of the hypothesis and presentation of objectives.

The results are exposed very clearly, with appropriate parameters, evaluation of significance clear and well expressed. The graphics are correct, of good quality and very illustrative of the idea they intend to convey.

The methodology is clear and well expressed.

The discussion is broadly correct. Even so, it must be considered somewhat extensive for the ideas it conveys, since it results in repeating concepts related to the pathophysiology of hypertension already mentioned in the introduction, and it fails to compare the results obtained with other models and similar publications, as well as in detailing the strengths and virtues of the work. In particular, the limitations must be detailed and developed, including the animal model, the lack of real clinical evidence, and the need for further studies.

The conclusions are clear, concise and very coherent. Assertions are made supported by the results presented.

As a final balance, the work presents a good academic level, contributes novel data to the understanding of experimental models of hypertension, and could be published after the abstract and discussion are improved, which constitutes minor but important changes.

I haven't comments.

Author Response

This is an experimental research work on animals using Two groups of male F2(ISIAHxWAG) hybrids of different ages. A statistical analysis of principal component data science is carried out, analyzing the relationships of morphological, physiological and behavioral traits. It has been shown that the development of stress-sensitive hypertension, accompanied by a persistent increase in basal blood pressure and an increase in the basal concentration of plasma corticosterone, is accompanied by a decrease in response of these parameters to stress and by an increase in anxiety. The study revealed age-dependent relationships between key features and their changes with age and worsening of the hypertensive state.

Answer: The authors thank the reviewer for the high evaluation of our work and comments, and suggestions that allowed us to improve the text of the manuscript. For the convenience of reviewer, all corrections made to the text of the manuscript are shown in red. As recommended by the reviewers, the text of the manuscript has been edited by a professional translator. The language certificate is enclosed.

Regarding the manuscript in detail:

The title is clear, concise, brief and well describes the work and model used.

Answer: Thank you

The abstract is clear, but it should be corrected to be structured, respect the different sections of the manuscript and provide some numerical data on results and levels of statistical significance.

Answer: According to the Instructions for Authors the abstract should follow the style of structured abstracts (Background, Methods, Results, Conclusion), but without headings. The authors have tried to present the Abstract in the required style. Thus, we did not introduce headings into the text, but made some changes to the text of the abstract so that it complied with the recommendations of the journal and the reviewer. In addition, according to the comment of the reviewer, the necessary information about the levels of statistical significance was included in the text of the Abstract (Lines 28-31).

The introduction is somewhat extensive, but it still meets the requirements of adequately explaining the theoretical foundations of the research, raising the doubts that lead to this work, and laying the foundations of the hypothesis and presentation of objectives.

Answer: Thank you

The results are exposed very clearly, with appropriate parameters, evaluation of significance clear and well expressed. The graphics are correct, of good quality and very illustrative of the idea they intend to convey.

Answer: Thank you

The methodology is clear and well expressed.

Answer: Thank you

The discussion is broadly correct. Even so, it must be considered somewhat extensive for the ideas it conveys, since it results in repeating concepts related to the pathophysiology of hypertension already mentioned in the introduction, and it fails to compare the results obtained with other models and similar publications, as well as in detailing the strengths and virtues of the work. In particular, the limitations must be detailed and developed, including the animal model, the lack of real clinical evidence, and the need for further studies.

Answer:  We have included in the text of the discussion related publication concerning hypertensive SHR rats (Lines 366-370), and also supplemented the discussion with text indicating the limitations of the model, as well as emphasizing the novelty and potential usefulness of the results for further studies in human populations (Lines 484-502).

The conclusions are clear, concise and very coherent. Assertions are made supported by the results presented.

Answer: Thank you

As a final balance, the work presents a good academic level, contributes novel data to the understanding of experimental models of hypertension, and could be published after the abstract and discussion are improved, which constitutes minor but important changes.

Answer: Thank you. The authors hope that the corrections and additions made to the text of the manuscript are sufficient and the corrected version of the manuscript is suitable for publication.

Reviewer 3 Report

The study by Oshchepkov et al. presents and in-depth discusses age-dependent changes in the key features involved in the pathophysiology of hypertension in ISIAH rat males. This study is generally well-written and brings interesting findings which may contribute to the development of novel hypertension´s treatment strategies. However, I have some minor remarks before I recommend the manuscript for publication.

1. As cortisol levels fluctuate throughout the day, the blood used for its evaluation should be taken at the same time during the day. I recommend the authors clearly state when the rats' blood was taken. A similar statement should be included for blood pressure measurements.

2. line 373 – the word „stress“ is written in a different font style or size

Author Response

The study by Oshchepkov et al. presents and in-depth discusses age-dependent changes in the key features involved in the pathophysiology of hypertension in ISIAH rat males. This study is generally well-written and brings interesting findings which may contribute to the development of novel hypertension treatment strategies.

Answer: The authors thank the reviewer for the high evaluation of our work and comments, and suggestions that allowed us to improve the text of the manuscript. For the convenience of reviewers, all corrections made to the text of the manuscript are shown in red. As recommended by the reviewers, the text of the manuscript has been edited by a professional translator. The language certificate is enclosed.

However, I have some minor remarks before I recommend the manuscript for publication.

  1. As cortisol levels fluctuate throughout the day, the blood used for its evaluation should be taken at the same time during the day. I recommend the authors clearly state when the rats' blood was taken. A similar statement should be included for blood pressure measurements.

Answer: In the text of subsection 4.2. (lines 536-538) information has been added to clarify the timing of the procedures for measuring the level of blood pressure and taking blood samples for the subsequent determination of the plasma concentration of corticosterone. The corrections made in the text of the manuscript are shown in red.

  1. line 373 – the word „stress“ is written in a different font style or size

Answer: corrected. Thank you.